# “We Are Having a Huge Problem with Compliance”: Exploring Preconception Care Utilization in South Africa

**DOI:** 10.3390/healthcare10061056

**Published:** 2022-06-06

**Authors:** Winifred Chinyere Ukoha, Ntombifikile Gloria Mtshali

**Affiliations:** School of Nursing and Public Health, College of Health Sciences, University of KwaZulu-Natal, Durban 4001, South Africa; mtshalin3@ukzn.ac.za

**Keywords:** preconception care services, healthcare utilization, preventive care, maternal and child health, social-ecological model

## Abstract

Background: Preconception care (PCC), a policy directive from the World Health Organisation (WHO), comprises all the health interventions offered to women and couples before conception and is intended to improve their overall health status and the pregnancy outcomes. Although PCC should be an essential part of maternal and child health services in most African countries, its provision and utilization are not widely documented. Hence, this study aimed to explore the factors influencing preconception care utilization among high-risk women in South Africa. Methods: A descriptive qualitative study of 29 purposively selected women and healthcare workers was conducted through individual in-depth interviews using a semi-structured interview guide. The interviews were transcribed verbatim, and the analyses were performed using Nvivo version 12. The Social-Ecological Model (SEM) guided the data analysis. Four levels of factors (the individual, the interpersonal, the community and social, and the policy and institutional) were used to assess what can influence PCC utilization. Findings: The availability of PCC services, the intrahospital referral of women, the referral practices of other healthcare workers, the underutilization of the PCC facility, and resources emerged at the institutional levels, while compliance with PCC appointments, socioeconomic factors, pregnancy planning, assumptions, and knowledge was at the individual levels. Conclusion: The utilization of the preconception care services was inadequate. The primary influencer of preconception care utilization was at the individual, policy, and institutional levels. The availability of preconception care services and the intrahospital referral of women at high risk of adverse pregnancy outcomes positively influenced the women’s PCC utilization, while poor pregnancy planning, and unavailability of PCC policies and guidelines negatively influenced preconception care utilization. Therefore, interventions to improve PCC utilization should focus on the four SEM levels for effectiveness. There is a need to raise PCC awareness and develop policy and guidelines to ensure consistent, standardized practice among healthcare workers.

## 1. Background

Preconception care, a policy directive from the WHO, comprises the biomedical, behavioural, and lifestyle health intervention provided to women and couples before conception and is intended to improve their health status by reducing social and environmental factors that could lead to poor pregnancy outcomes [1]. PCC services can be in the form of preventive, promotive, curative, and social intervention provided before pregnancy, or they can be interventions in the period extending from three months before and interventions provided between two pregnancies [1]. Although PCC should be an essential part of maternal and child health services in most African countries, its provision and utilization are not widely documented [2]. Globally, the utilization of PCC in developed countries is picking up momentum, with many European countries having policies, guidelines, and recommendations for PCC [3], but this is not always the case for most developing countries where guidelines are still not available [4]. In meeting the sustainable development goal (SDG) number three, which aims to reduce maternal and new-born mortality to 70 per 100,000 and 12 per 1000 live births, respectively [5], preventive care such as PCC could have a significant impact. Several strategies have effectively promoted PCC uptake among the general population, such as community outreach programs in which women at high risk of perinatal morbidity and mortality are given information about PCC or are invited for PCC consultation [6].

In most African countries, the level of utilization of PCC in the general population and among women at high risk of adverse pregnancy outcomes is relatively unknown [7]. PCC’s utilization among women with chronic medical conditions, such as rheumatic heart disease and diabetes in Sudan [8], Zambia, and Ethiopia [9,10], revealed a suboptimal utilization among this vulnerable group of women. HIV testing and counselling were the most utilized PCC components in many African countries [7,11,12], while folic acid supplementation was among the least utilized components [12,13,14]. The reasons for the non-utilization of PCC services range from lack of awareness of its availability and importance [15,16,17] to non-accessibility of the service and the cost implication [16].

A study on the utilization of maternal health care services among Nigerian women revealed that most women were aware of the general maternal services available but were unaware of the extent of the services provided. It further identified factors associated with the non-utilization of services, such as availability, acceptability, quality, affordability, and individual characteristics [18]. In other studies, the identified barriers to the utilization and provision of PCC services were unplanned pregnancies and lack of awareness among women [4,19,20]. In Ethiopia, women’s utilization of PCC was low, while the availability of PCC units, knowledge about PCC services, level of education, marital status, and age were among the factors that affected the uptake of PCC [11]. A systematic review revealed that women’s utilization of PCC was significantly low at 16.27% and demonstrated that age and proper knowledge of preconception care showed an association with utilization [21].

Although PCC utilization remains low globally, pregnancy intent and previous adverse pregnancy outcomes are robust indicators of PCC use; therefore, strategies to enhance access to PCC should incorporate ways to ensure planned pregnancies [22]. Other unknown factors affect women’s utilization of inter-conception care; these factors will not assure the long-term and consistent utilization of preventive care among vulnerable women even after removing the common barriers to care [23]. Contraceptive use and the prevention of unintended pregnancy have been viewed as essential factors that enhance the utilization of PCC and counselling. Thus, the need for preconception contraceptive counselling increased with childbirth postponement, and this provides a window of opportunity for the improvement of health status by stressing the health benefits of PCC [24].

A study in the Netherlands on the determinants of the intentions to use PCC revealed that working-class women were less likely to consult for PCC; thus, it was recommended that health care professionals revisit their schedule and possibly introduce evening consulting hours for convenience [25]. The centralized guideline for the provision of standardized PCC in China was said to be not suited for the entire population as the healthcare needs of the local population were not considered; therefore, for a more significant impact on reproductive health, both at the national and the regional level, some modification was indicated to be necessary [26]. An Australian study among pregnant women revealed that about 8.3% used complementary and alternative medicine during the preconception period to enhance their health instead of seeking biomedical services such as PCC [27].

There is still a paucity of studies on the utilization of PCC in South Africa. However, there have been lessons learned from a few studies on the continent. Although many of the studies used quantitative method, most of these countries do not have dedicated preconception care clinics. Thus, the need to hear the voices of the women and healthcare workers (HCWs) in this unit concerning service utilization. Therefore, this study explored the factors influencing preconception care utilization among high-risk women in South Africa.

## 2. Materials and Methods

### 2.1. Study Design and Setting

A qualitative descriptive design was used to permit a deeper understanding of preconception care utilization [28,29]. This study was necessary to provide a more in-depth description of the viewpoint of the women and healthcare workers with regard to preconception care utilization.

The current study was conducted in a referral tertiary hospital with a dedicated pre-pregnancy clinic in eThekwini Metropolitan Municipality, in the KwaZulu-Natal province of South Africa. This hospital is a tertiary referral hospital that only attends to critical and high-risk medical and surgical conditions. eThekwini Metropolitan Municipality is among the biggest municipalities in South Africa, with an estimated 3.4 million inhabitants, and is located on the east coast of South Africa [30]. The municipality covers an area of about 2297 km^2^, according to the census of 2011 [31]. The study was conducted in the obstetric and gynaecological unit, which manages women at high risk of adverse pregnancy outcome and obtains referrals from all over the province and beyond. This setting was deemed adequate on the premise that the population of healthcare workers and women attending services in this clinic were better informed about PCC service utilization. The obstetric and gynaecological unit has obstetric and gynaecological, feto-maternal, genetic, and pre-pregnancy clinics. The services offered include preconception and genetic counselling, screening services, management of chronic conditions, treatment, and care for women of childbearing age.

### 2.2. Population, Recruitment, and Sampling

Women of childbearing age (18–49 years) at high risk of adverse pregnancy outcomes due to their medical or surgical conditions and the healthcare workers who provide care to these women in the selected preconception care clinic were the target population for this study. The choice of the population was on the premise that they would be able to provide the required information on preconception care utilization as most of them have or should have attended preconception care services in the past. This unit attends to approximately 80 clients per week in the four clinics that run in the unit. This unit provides services mainly for women, but occasionally, some men are seen alongside their partners during genetic screening and pre-pregnancy counselling. The inclusion criteria for patients are that they must be (1) an adult patient within the childbearing age, (2) a patient with a medical or surgical condition that should necessitate a preconception care service, or (3) a healthcare worker who has worked in this clinic for more than a year.

Before collecting data, gatekeeper permission to access the clinic and the selected participants was granted by the managers responsible for the clinic. The researchers visited the clinic to introduce the research objectives and aims and to request participation and identify potential participants. The healthcare workers assisted in pointing out the potential participants to be recruited from the preconception care and the genetic, obstetric, and gynecological clinic and ward. Non-probability purposive sampling was used to select 24 women and 5 healthcare workers for this study, and the sample size was based on data redundancy.

### 2.3. Data Collection

The data collection and initial analysis were conducted concurrently between October and December 2020. The concurrent process of data collection enabled the determination of the point of data redundancy. Data saturation/redundancy was when no new information emerged from the interviews and the data collection should be concluded. The data collection for this face-to-face in-depth individual interview was carried out using a semi-structured interview guide after content validation of the instrument by experts in qualitative research. The interviews were conducted at a preferred location of the participants within the hospital. The interview questions explored the participant’s knowledge regarding preconception care. For women who were unaware of the meaning of preconception care, a further explanation of the concept was provided. Their experience with preconception care utilization was further assessed; this included the challenges to PCC utilization. The participants were asked the same questions, and varying probing questions were used to obtain clarifications where necessary. Field notes were also taken during the data collection while observing the setting. The data were collected in English or Zulu, the local language, based on participant choice by the first author and another doctorate nursing student with previous experience in qualitative data collection. The research assistant was acquainted with the interview guide and the concept in a two-day training session before the data collection. The interviews were audio-recorded and lasted for 25 to 60 min with few individual differences.

### 2.4. Data Analysis and Theoretical Framework

The data were analysed using Nvivo version 12 software and the Social-Ecological Model (SEM) for the initial coding guide. The SEM proposes a combination of multilevel factors that interplay to influence individual health behaviour, including individual, interpersonal, organizational, community, and public policy [32]. Adopting this model can help to provide a broad perspective of the factors influencing health behaviours in order to develop an effective multifaceted population health intervention towards health improvement [33]. The SEM is appropriate when ensuring that environmental and personal factors are considered when analysing health behaviours [32]. Therefore, using the SEM to explore the factors influencing women’s utilization of PCC enabled the integration of the four levels of factors necessary for effective intervention because motivating individuals to change their behaviour cannot be effective if the environment and the policy are inadequate. The data were coded by two independent coders guided by the SEM and the research objectives. The initial coding framework was generated, and then, the following emerging codes were inductively added to form the model of factors influencing PCC utilization (Figure 1). The emerging codes were scrutinized and validated by the research team. The consolidated criteria for reporting the qualitative research (COREQ) checklist were used to guide the writing of this qualitative study [34].

### 2.5. Trustworthiness

Trustworthiness is a concept of ensuring authenticity and quality in a qualitative study. As proposed by Lincoln and Guba [35], trustworthiness criteria were followed to ensure the accuracy of the study findings; these include credibility, dependability, confirmability, and transferability. Credibility, which resembles validity in a quantitative study [36], was ensured in this study by the participants’ and the co-investigators’ validation of the study findings. Dependability, which is close to reliability, was ensured by keeping track of the coding decisions for the stability of the data and by involving more than one researcher in the data analysis [37]. Transferability, which corresponds to generalization, was ensured by using a representative sample in the study and a detailed description of the context of the study to enable study replication. Confirmability was ensured through the objectivity and neutrality of the study findings.

### 2.6. Findings

A total of twenty-nine in-depth interviews were conducted with women and health care workers (24 women and 5 healthcare workers). Among the five HCWs, two specialized in genetics, one in obstetrics and gynaecology, and one in fetal medicine, and one was a family planning nurse. Among the women in the study, fourteen were pregnant while ten were not pregnant, and ten among the fourteen pregnant participants had cardiac conditions. Their ages ranged between 20 and 45 years (Table 1).

### 2.7. Using the Social-Ecological Model to Explore PCC Utilization

The SEM layers were used to group the emerging factors from the study findings into the individual, interpersonal, community and social, institutional, and policy-level factors influencing women’s utilization of PCC (see Figure 1). Information on the institutional and policy-level factors was supplied by the HCWs only, while both the HCWs and the women supplied the information on the remaining three levels.

### 2.8. Individual-Level Factors

Among the 24 women who participated in the study, only 4 had attended a tertiary institution but only at a diploma level. The majority, 79% of the women, had only completed the secondary level of education, and one had only finished primary education. There was limited understanding of maternal and child health services among the participants whose highest educational qualifications were secondary education and below. The emergent factors that were included under the individual-level factors were compliance with PCC appointments, non-utilization of PCC services, socioeconomic factors, pregnancy planning, assumptions, knowledge, and attitudes.

### 2.9. Poor Compliance with PCC Appointments

Most women with severe health conditions are usually counselled about their condition and advised to visit the pre-pregnancy clinic for assessment and screening when planning to fall pregnant. The screening is necessary to rule out any conditions that may affect the pregnancy outcome and to ensure that they start pregnancy at optimal health.

“We advise our patients even on discharge that before you decide to fall pregnant again, please come and see us in our pre-pregnancy clinic … we tell them it is documented in their discharge summaries … we are big on PCC, but I don’t think it happens everywhere”.(HC3)

Nine out of ten women with cardiac conditions who received PCC information in their previous pregnancy or during their cardiac surgery indicated that they did not comply with the advice given to them by the healthcare workers. One of the reasons given by participants for non-adherence was poor pregnancy planning.

“The doctor told me I must come to the hospital to discuss it if I plan to fall pregnant because the drugs, I am taking are so dangerous to the child. So that the doctor can control the drugs, but I never come to the hospital because I didn’t plan to fall pregnant until I noticed I was four months pregnant”.(PC1)

“I was informed that when I am about to fall pregnant, I should come because I have got a heart problem so that they can discuss what I am going to do. I did not come back to the hospital before I became pregnant. I did not follow what they told me, but when I realize that I was pregnant, I did stop the warfarin. I didn’t plan to fall pregnant. It just happened”.(PC2)

Three healthcare workers further reiterated the non-adherence to PCC appointments by the patients to be a significant issue affecting PCC utilization because of lack of pregnancy planning. They are more likely to comply with PCC appointments if it is for genetic conditions than for other maternal health issues, and the genetic nurses stated that some patients would come for the appointments.

“We are having a huge problem with compliance; I would say one-third of the patients don’t pitch after an appointment is given. A lot wouldn’t come, yet some would come to get more information because they want to plan another baby. some will not pitch because they don’t see the relevance, but I would like to see the number increasing”.(HC1)

“We do see patients who are told by us on discharge after delivery to come for PCC, who have not who have actually fallen pregnant and then come to me, the reason is that many pregnancies are unplanned”.(HC3)

### 2.10. Contraceptive and Folic Acid Utilization Experience among Women

Contraception and family planning are part of preconception care strategies that ensures that women have a planned pregnancy. Most women at high risk of adverse pregnancy outcomes were not using any form of contraception. They are usually placed on short-acting contraceptives, but they rarely continue them for long. None of the participants indicated that they had used a long-acting contraceptive before.

“I have used contraceptives before. I was on the three months injection, Depo-Provera … I stopped because I was bleeding and having an allergic reaction, it itches a lot”.(PC1)

“I was on Traphasil 2009 to 2011, but I defaulted I didn’t have any reason, I just didn’t want those pills anymore”.(PC21)

All 14 pregnant participants, including those with genetic conditions, had not used folic acid preconceptionally. They were all placed on folic acid after pregnancy was confirmed. They stated that they were unaware of the benefits of periconceptional folic acid use and had not been informed about it.

“The doctor gave me folic acid after I became pregnant. I was on folic acid during pregnancy, not before pregnancy”.(PC14)

“… no one informed me that I should take folic acid before pregnancy, I did not know about it, I started taking folic acid after I became pregnant”.(PC13)

Some women did not even know folic acid by its name, let alone its indication, but only through the description.

“What is folic acid? … No, I didn’t take anything, I didn’t take folic acid before pregnancy …”(PC23)

### 2.11. Unplanned Pregnancy

Both the healthcare workers and the women admitted that the main reason for the non- and under-utilization of PCC services is due to unplanned pregnancy among women. The healthcare workers described unplanned pregnancy as an enormous global problem that needs to be addressed.

“Three-quarters of our pregnancies are unplanned. In fact, in KwaZulu-Natal, 70% of our patients come as unplanned or unintended pregnancies, so they do not come to you in time. They are coming to you when they are already pregnant”.(HC1)

“Most patients reason for not seeking PCC is that pregnancy was just unplanned, they have not been on contraception. This is not a challenge for local only, is a worldwide challenge … majority of the pregnancy are unplanned”.(HC3)

Most of the participants confirmed the issue of unplanned pregnancies by indicating that their pregnancies were not planned but were accepted. Interestingly some of them stated that they were surprised to find out that they were pregnant even when they were not on any form of contraception. Very few pregnancies were planned, and only one participant stated that she fell pregnant while using the (oral) contraceptive.

“It was not my plan to fall pregnant, it just happened, and I was shocked when I noticed that I was already four months gone”.(PC1)

“I was on Traphasil, the tablet … then well it didn’t work properly I don’t know why it didn’t work I can say due to stress I don’t know”.(PC13)

### 2.12. Women’s Attitude towards PCC Information

Three women with cardiac conditions indicated that they did not understand the information given to them during PCC counselling clearly, and that they were sometimes not given the correct information, but they always pretended to know it all. This limited understanding of what PCC involves hampered their utilization of the services and adherence to PCC appointments.

“Most of the time, we women think we understand everything, and we don’t need more information of which we have the wrong information. If we take time to find information… understand and use it things will be better”.(PC3)

“… I will rather say we don’t understand the information given to us like most of us don’t follow the instructions because we don’t understand it like seriously, we don’t understand … most of us even if you can tell us about something, we don’t want to know more. Yes, it is our lack of knowledge, sometimes we don’t understand that language (medical language), and we have already given up”.(PC20)

### 2.13. Assumptions Made by Women about Healthcare Workers and PCC

The women with cardiac conditions indicated some assumptions about healthcare workers which had prevented them from seeking PCC. Some of those assumptions ranged from the HCWs not wanting them to have more babies due to their condition to the HCWs shouting at them if they indicated their pregnancy intention, and some saw being unmarried as a barrier to seeking PCC. It also highlighted the danger of values imposed by healthcare workers.

“It is not right we cannot be able to tell the nurses that we want to fall pregnant because they use to tell us that as we are cardiac patients, we are not allowed to fall pregnant now and again. Therefore, it is easier for me to come here already pregnant … it is better when we are already married because the nurses are always complaining about everything. I should not ask for PCC”.(PC15)

“The sisters told me to come when I plan to fall pregnant … I did not come for PCC before I fell pregnant because the nurses will shout at me if I come to tell them that I would want to fall pregnant”.(PC18)

### 2.14. Socioeconomic Factors

The participants indicated that sometimes women do not adhere to PCC appointments due to financial reasons and lack of money for transport as most of them are not employed.

“… there are patient’s issues where patients don’t have resources to get to the hospital because of lack of transport, and a lot of them are not working”.(HC3)

“… sometimes patients just did not come for PCC due to financial thing the hospital or clinic is too far for them to get to”.(HC1)

### 2.15. Interpersonal Level Factors

The identified interpersonal factors related to partner support that could affect women’s PCC utilization.

### 2.16. Women’s Partners’ Influence

The healthcare workers also opined that the women’s partners contribute to the under-utilization of PCC services. This is because of their absence during the counselling session; they do not realize the need for pre-pregnancy appointments.

“Sometimes you find that the challenge is not from the patients but from the partner who doesn’t understand why she needs to come to the hospital now that she not pregnant because PCC is not a concept that everybody is aware of”.(HC3)

The participants’ partners have a significant influence on their utilization of PCC. A participant indicated that the partner could not make the time to come with her for the PCC appointment because he was working. For that reason, she defaulted on the PCC appointment.

“For me, my man got no time … … he is working and could not get time off work, and we are supposed to come together”.(PC22)

### 2.17. Community and Social Level Factors

Culture and belief were the only emerged factors under the community and social level of influence on PCC utilization.

### 2.18. Culture and Belief

The participant’s culture also influenced their utilization of PCC services when planning to fall pregnant. This is related to the aspect that has to do with medical interventions and revealing one’s pregnancy intentions to an outsider as they believe this may negatively affect the pregnancy outcome. Another participant stated that she does not like contraceptives and does not use them because of her belief that if she is meant to fall pregnant, that nothing will stop it.

“… some culture doesn’t believe in English medicine and involving those things when one is planning pregnancy. But I think that every woman needs to take care of their life. Like in my culture, we believe that you can have a miscarriage if you tell someone that you want to fall pregnant because of this I don’t think I can come for PCC”.(PC20)

“I have never used contraceptives before … I don’t like it, and I don’t believe that I have to use contraceptives. I don’t know if it might happen for me to have a baby; it will happen; I can’t stop it anyway … it is just what I believe in”.(PC23)

### 2.19. Institutional and Policy Level Factors

We included the availability of PCC services, the intrahospital referral of women, the referral practices of other HCWs, the under-utilization of the PCC clinic, resources, and policy-related factors under the institutional and policy factors influencing PCC utilization.

### 2.20. Availability of PCC Services and Intrahospital Referral of Women

Various PCC services are available in the selected hospital, but the healthcare workers’ practices shape the use of these services. Most of the patients seen for PCC and counselling come from within the hospital because the most complicated medical and surgical conditions are managed there.

“We have the dedicate pre-pregnancy clinic here, so patients are referred to us from the obstetric clinic here, the high-risk clinic, the fetal-medicine clinic, and on few occasions, we normally get outbounds coming in from another hospital”.(HC1)

“… we are usually asked to review cardiac patients regarding a potential pregnancy … we do have in-hospital referral because loads of women with many severe conditions are managed in this hospital”.(HC2)

Some patients with chronic conditions are also referred from within the hospital for preconception care counselling before they are discharged after delivery. Most of them are placed on birth control to ensure planned pregnancy.

“… we see patients with serious medical conditions, who have delivered their babies here, so before discharge, we counsel them about family planning and place them on appropriate contraceptives”.(HC4)

### 2.21. PCC Referrals and Screening from the Base Hospitals

However, HCWs reiterated the poor PCC referrals and screening practices of other health workers in the base and district hospitals as a considerable problem. They indicate that women are often referred for PCC after pregnancy has occurred.

“I do get calls from other institutions for patients with losses and patients with medical problems. They do get referred occasionally from the base hospitals but not in the kind of numbers that I would like to see because I see patients after they have fallen pregnant, which is sometimes a bit late”.(HC3)

“… if people don’t refer, then we will not see enough patients, we get few referrals from the base hospitals for the pre-pregnancy care”.(HC1)

### 2.22. Under-Utilization of PCC Clinic

Despite the enormous benefits of PCC, the dedicated PCC clinic in the hospital is under-utilized by the women for various reasons. The HCWs indicated that they would like more women referred for PCC and other clinic and base hospital professionals to improve their referral practices. The main reasons cited for the under-utilization of the dedicated PCC clinic are the inadequate PCC screening and counselling and poor referral by HCWs.

“There are huge benefits of PCC, and I would say that our PCC clinic should be more utilized. Unfortunately, our PCC clinic is under-utilized because people are not in the habit of referring patients to the clinic … it will make a huge difference if patients are referred. Right now, there are months I might not see any patients”.(HC3)

“… the problem with the clinic is that it is under-utilized because HCWs in the other disciplines do not identify women adequately, do not counsel women about fertility issues, so we actually lose the opportunity to refer these patients to our clinic so that they can be accessed before pregnancy”.(HC4)

### 2.23. PCC Material and Resources

Material resources are required for the adequate provision and utilization of PCC services. These materials will remind women of the importance of pre-pregnancy visits and the need to be in an optimal state of health before conception. There is a lack of PCC technology-supported applications compared to what is seen in other health services such as ante-natal care. There is a lack of PCC posters, pamphlets, and internet services that participants think will assist in disseminating information about PCC.

“Why can’t we have something like mom connect (a WhatsApp application for pregnant women) for non-pregnant women … everybody got a smartphone, and we have WhatsApp, messages will be sent about the importance of PCC”.(HC3)

“… am not aware of any pamphlets, posters or any other PCC materials, we need those, we don’t have access to the internet so if you want to give a patient literature to take home that they can read and exchange with their family is very difficult for us to do that”.(HC5)

They further reiterated the poor awareness level among the HCWs regarding the need to see patients before conception because of non-prioritization and inadequate training. They further disclosed poor contraceptive knowledge among HCWs outside obstetric and gynaecological units as they only emphasize the contraindication of contraceptives to women without making efforts to verify what can be indicated for those conditions. For this reason, women see contraceptives as contraindicated for their conditions without any knowledge of which contraceptives they can use.

“There is a lack of knowledge from the health care worker perspective about the importance of seeing patients preconceptionally … doctors in particular and nurses are not fully aware of the value of PCC because they think they have bigger problems. So, we think that PCC is nothing important, but I think it is a big mistake … there is no PCC in-service education, workshops, and training, PCC is not given enough emphasis”.(HC3)

“… health workers outside of obstetrics and gynaecology poorly understood contraception, they only know contraindication to some of the contraception. For example, in the rheumatology clinic, a patient may be told you can’t use the combined oral contraceptive because you are at risk for venous problems. We stop there but shouldn’t we tell the patient what they can do, we are telling them what they can’t use, but we are not telling them what they can use”.(HC2)

### 2.24. PCC Policy and Guidelines

There are no policy guidelines for PCC provision at either the national or the local department levels. These guidelines should enable the standardized provision of PCC services at all healthcare levels and remind HCWs about the need to see women preconceptionally.

“We have national guidelines on how to treat HBP in pregnancy. We need something similar. PCC directives should come from higher up so that people will have to do it. There are no policy and guidelines, and we need one”.(HC3)

“There isn’t a PCC guideline, nationally or from the local department of health, so non am not aware of any … but there is the maternity care guideline which is a South African guideline that is made for Primary Health Care, it does mention the issue of PCC, but that is not sufficient”.(HC2)

## 3. Discussion

The individual interviews conducted amongst the women and healthcare workers revealed several interlinked factors, particularly the individual and institutional and policy-related factors that influence women’s utilization of PCC in South Africa.

Women with previous heart surgery were given PCC advice and appointments in their previous pregnancy, but none complied. However, they were fully aware that they required proper assessment of their heart condition before pregnancy and that their blood thinners needed to be changed immediately after pregnancy was confirmed, due to their teratogenic nature, but could not do so for various reasons. Most women only visited the clinic a few months after conception to have their medication changed and some as late as four months. The preconception health advice among women in this study was higher than that among women with rheumatic heart disease, which revealed that more than half of them were not counselled about becoming pregnant [8]. Good knowledge of PCC services and concepts and having a chronic condition all increase PCC services utilization [38], but in this study, the knowledge of the women about PCC did not increase their utilizations because of other reasons. Contraceptive counselling is a vital PCC strategy among women, especially those at risk of adverse pregnancy outcomes, to ensure planned pregnancy [1]. Contraceptive services were not well utilized by most of the participants in the current study. Some stated that they disliked contraceptives and, therefore, were not on any, while others just defaulted for little or no reason. In the study setting, contraceptive services are primarily provided to women after childbirth, and they are responsible for its continuation in their local clinics and hospitals, but few continue. Our findings affirm a previous study among women with rheumatic heart disease, where more than half of the participants did not use contraception [8].

Furthermore, all the pregnant study participants did not use folic acid preconceptionally because they were unaware of its importance and indication. Interestingly, some of them currently on folic acid do not know it by name but could only recognize it through the description. The current study’s findings are in line with a study from Ethiopia where the utilization of folic acid before pregnancy was reported to be zero percent among women [14] and among Italian women who were also unaware of the indications for folic acid [39].

Both the HCWs and the women confirmed a lack of pregnancy planning as one of the influencers of PCC utilization. An estimated 70% of pregnancies were unintended. Studies have reported poor pregnancy planning in both developed and developing countries as one of the factors influencing the utilization of PCC services [19,20,40]. Furthermore, the participants believed that lack of understanding and knowledge of the PCC concept influenced the utilization of PCC services. Thus, this resulted in non-adherence with PCC appointments among women who were counselled on the need for PCC in their previous pregnancy. Ayele et al. revealed that Ethiopian women with good knowledge of provided PCC services are four times more likely to utilize PCC services than those with insufficient knowledge [21]. Women also perceived the healthcare workers as an impediment to their utilization of PCC services as they assume that they would not be supportive of their pregnancy intentions, and they would therefore rather not come for PCC services, while others assumed that PCC was not for single women and thus would be better for married women. These findings point to the need for preconception care services provided in a supportive and non-judgmental environment while ensuring that adequate information is given to women. This corroborates the findings of previous studies among women with diabetes which revealed that they were scared that the HCWs would discourage them from becoming pregnant because of their condition [9]. Women with rheumatic heart disease were indifferent about PCC for their subsequent pregnancy even after an explanation about the concept and its benefits was given [8]. In contrast, among women with pre-existing diabetes, some did not believe that PCC was necessary [9]. Likewise, socioeconomic factors were identified as PCC utilization influencers as some women lacked the financial resources to get to the clinic. Among Kenyan women, cost implications also prevented them from seeking PCC [12].

Under the interpersonal level factors, the partner’s support strongly emerged as the factor influencing women’s PCC utilization. Most women could not adhere to their PCC appointments because their partners did not understand why they should visit the health facilities before conception due to their absence during the counselling session and their poor PCC awareness. The joint PCC discussion with the partner increased PCC knowledge [41], while the husband’s support was associated with increased PCC utilization [38].

Some cultural and belief influences were further identified at a community and social level as some women assert that other cultural beliefs prohibited them from telling anyone, mainly an outsider, about their pregnancy intentions. This they believe may negatively affect the pregnancy outcome if they do so. The women’s preference to keep pregnancy planning a secret has been revealed as a hindrance to PCC utilization among ethnically diverse women in the UK [42] and among Italian women who maintained that pregnancy intension should be a natural event [39].

Several factors also influenced PCC utilization at the institutional and policy level, such as inadequate screening and referral of the patients from the PHC clinics, as most patients seen for PCC services are usually referred from within the study institution. Most high-risk patients are referred by the base hospital when they are already pregnant, when PCC interventions are too late for the current pregnancy. An Australian study also identified a lack of practitioners’ referral as being due to poor PCC consultations among women with diabetes [43]. Ideally, all hospitals and every PHC facility should provide PCC services, and they should also be core services in every family planning unit [44]. However, we found that this essential preventive service is widely neglected, and many women start their pregnancy journey without screening and primary preventive care. From the perspective of the HCWs, the dedicated PCC clinic in the current study is under-utilized due to the inadequate screening and referral practices of the healthcare workers in the PHC clinics and women’s poor pregnancy planning practices. A dedicated preconception care unit has been recommended [19], but this is not a norm for many low- and middle-income countries where access to maternal and child health services is still the priority. However, this luxury has been neglected and under-utilized in South Africa.

Furthermore, the availability of policy and guidelines will enhance PCC provision and ultimately lead to PCC utilization [45]. Many developed countries in Europe have a PCC policy and guidelines for women at high risk of adverse pregnancy outcomes [3,46]. HCWs stressed the lack of PCC policy and guidelines, which will ensure a standardized method of PCC provision and act as a reminder of the importance of seeing women before pregnancy. Inadequate material resources were also identified at the institutional level as HCWs did not have access to PCC posters, checklists, and pamphlets. These materials will assist in raising awareness about PCC and available services. The availability of PCC posters and checklists were identified as facilitators of PCC utilization [47]. A study among primary health care nurses in South Africa revealed a lack of PCC resources [48]. Finally, insufficient awareness, skills, and knowledge regarding the importance of PCC among HCWs were identified as influencers of PCC utilization and provision. Unfamiliarity among HCWs about the need for inter-conception care was an identified barrier to inter-conception care provision in the Netherlands [49].

It is essential to pay attention to both the environmental and the personal factors. Therefore, as Golden and Earp [50] suggested, multilevel interventions are necessary to address all the identified factors influencing PCC utilization among women.

### Strengths and Limitations of the Study

To the best of our knowledge, this is the first paper that considers the perspectives of women and healthcare workers with regard to preconception care utilization in a dedicated PCC service in South Africa. Using the SEM for data analysis helped to reveal areas where interventions should target for the improvement of PCC utilization. The study was part of a more extensive study on the PCC provision in public health facilities in the country, and in this exploratory study, two researchers collected and analysed the data to avoid bias from personal interest. Most studies on PCC utilization in Africa utilized a quantitative data collection method; therefore, the current research has filled the identified methodological gap. The main limitation is that the study was carried out in a tertiary institution with a dedicated PCC clinic; the institution obtains referrals from all over the province. However, the women commonly referred to this facility for PCC are those with chronic or genetic conditions and those at high risk of adverse pregnancy outcomes. Therefore, the findings may not represent the PCC utilization of the general population but may give a clue in that regard. The study also involved a small group of participants, and the findings from this group should be used with caution. Another limitation is that PCC is for men and women, but in the current study, only women participated because the unit in which the data were collected caters mainly for women.

## 4. Conclusions

This qualitative study aimed to explore the factors influencing preconception care utilization among women. The findings provide insight into how all the four levels of elements in the SEM shaped the utilization of preconception care among women in South Africa. It was discovered that the primary influencer of preconception care utilization was at the individual, policy, and institutional levels with few interpersonal and community-level factors. The utilization of preconception care services and women’s compliance with PCC appointments in this study were inadequate from the viewpoints of the women and the HCWs. The availability of PCC services and the intrahospital referral of women at a high risk of adverse pregnancy outcomes positively influence women’s PCC utilization. In contrast, many other identified factors influenced PCC utilization negatively. The presence of many negative influencers, both on the individual and the institution levels, such as inadequate screening and referral practices among HCWs in the PHC clinics, poor pregnancy planning, insufficient level of knowledge, assumptions about HCWs, and poor compliance with PCC appointments among women, led to the under-utilization of the dedicated PCC clinic. The unavailability of PCC policy and guidelines, including other resources such as pamphlets and posters, also negatively influences PCC utilization. Therefore, strategies to increase PCC awareness among the public and increase pregnancy planning should be developed.

## Figures and Tables

**Figure 1 healthcare-10-01056-f001:**
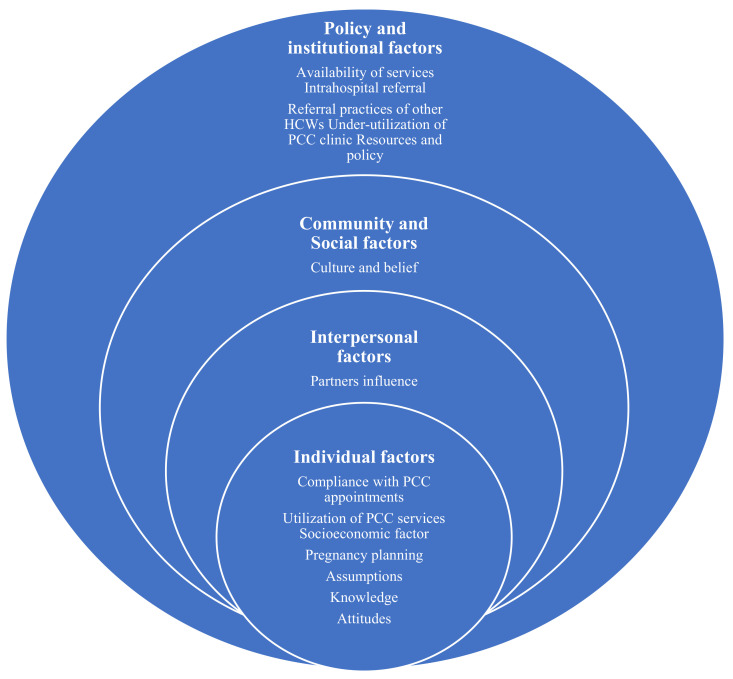
Factors influencing women’s utilization of preconception care based on the adapted Social-Ecological Model.

**Table 1 healthcare-10-01056-t001:** Demographic profile of patients (*n* = 24).

Pseudonym	Diagnosis	Age	Highest Education Level
PCI	Cardiac abnormality	32 years	Grade 12
PC2	Cardiac abnormality	22 years	Grade 12
PC3	Infertility	26 years	Diploma
PC4	Infertility	39 years	Grade 10
PC5	Infertility and HIV	32 years	Grade 12
PC6	Infertility and chronic anaemia	40 years	Bachelors degree
PC7	Cardiac abnormality, Obesity, and Hypertension	23 years	Grade 11
PC8	Cardiac surgery, diabetes, and Hyperthyroidism	33 years	Grade 12
PC9	Infertility	30 years	Grade 12
PC10	Infertility	45 years	Grade 11
PC11	Infertility	23 years	Grade 12
PC12	Infertility	30 years	Grade 12
PC13	Chromosomal abnormality	37 years	Grade 12
PC14	Chromosomal abnormality	30 years	Grade 10
PC15	Cardiac abnormality	26 years	Grade 12
PC16	Cardiac abnormality	26 years	Diploma
PC17	Kidney disease and Hypertension	29 years	Grade 12
PC18	Cardiac abnormality	20 years	Grade 10
PC19	Diabetes and Placenta previa	35 years	Grade 12
PC20	Cardiac abnormality and diabetes	39 years	Grade 12
PC21	Obesity	33 years	Grade 12
PC22	Cardiac abnormality and Hypertension	30 years	Grade 7
PC23	Cardiac abnormality and Obesity	29 years	Grade 12
PC24	Hypertension and Obesity	34 years	Diploma

## Data Availability

Data from this study is the property of the University of KwaZulu-Natal and may be made available upon request from the university or the study authors.

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
