# Peer review of "“We Are Having a Huge Problem with Compliance”: Exploring Preconception Care Utilization in South Africa"

_healthcare, 2022, doi:10.3390/healthcare10061056_

Round 1
Reviewer 1 Report
Excellent work, on a very relevant topic, especially in countries with a high birth rate.
This is an exploratory descriptive qualitative study, using the Social Ecological Model, on a very topical topic such as preconception evaluation, a very important topic to be able to guarantee the best postnatal results, especially in cases in which the mother has comorbidity before gestation.
The qualitative study is well planned and carried out. The Nvivo tool was used in its latest version, which is very suitable for coding the information and working on the saturation and redundancy criteria that are so relevant in this type of study.
Perhaps the intentional selection of cases is a relevant bias, although it is common in this type of study due to recruitment problems.
The application of the model is adequate and both the discussion and the conclusions are consistent with the study and its objectives.
The bibliography is relatively current, but pertinent to the study. In the case of WHO (1,2) or UN (5) references would include web access if available. Some reference is missing the full reference or web path (4,10, 12) .
As the authors acknowledge, perhaps the most important limitation is related to the fact that the study cannot be extrapolated to the normal population, since, being exploratory, it has focused on the population at risk.
I do not believe any specific correction to the study is necessary, given its exploratory nature.
Author Response
REVIEWER ONE

Reviewer 2 Report
“We are having a huge problem with compliance”: exploring 2 preconception care utilization in South Africa.
The adherence to preconception care and recommendations remains an issue in many areas of the world. The study of PCC-related issues was based on interviews of both patients, as well as healthcare workers, and highlighted some of the issues faced by these two categories with regard to PCC. Whereas the subject is interesting and relevant to clinicians, I have some observations:
- Some paragraphs are too long (ex: lines 52-71 or 72-91). Please split such long paragraphs into 2 smaller ones for an easier read;
- L284: “they were shocked to find out that they were pregnant” – please rephrase (ex: “expressed surprise”);
- Please modify figure 1 so that all the text can be seen clearly. Also, please change the shape (redesign), the figure is too large for its content.
Author Response
Thanks for the comments. Please find the attached document.

Reviewer 3 Report
The review of manuscript entitled “We are having a huge problem with compliance”: exploring 2 preconception care utilization in South Africa”
Tthe publication deals with a very important and current problem of preconception care. These issues are important not only in developing countries but also in highly developed countries, including Europe. This care is not always implemented correctly in rich countries and it can be expected that in poorer regions of the world it will also deviate from the standards proposed by the WHO.
The main goal of this procedure is to significantly reduce infant mortality and therefore should be widely and correctly used. Unfortunately, in many regions of the world this is completely impossible. The reasons are various, but most often they are related to the low economic status of the society, poverty, poor organization of health care and poor education of the society.
Therefore, it can be assumed in advance that the results of this publication will show negative results. There is simply no other option. Is it unknown knowledge, or can the authors propose solutions?
Therefore, the following questions should be asked, is this publication really needed? does it provide us with new and interesting information?
Unfortunately, not in my opinion. And this could be a shortened review of the manuscript.
However, my specific comments are as follows:
1. study was conducted in a referral tertiary hospital that only attends to high-risk obstetrical conditions, it seems to me that the choice of such a center does not reflect the level of medical care in any region of Africa. It can be assumed that the results from other medical centers will be even worse
2. women> 18 years of age were qualified for the study, while it is known that in this region of the world a huge percentage of mothers are underage - such a selection of the tested sample is associated with a large error
3. caring for about 80 patients per week shows little experience of the center
4. only 29 women were enrolled in the study, which is dramatically low. No conclusions can be drawn from such a small sample
5. the authors very briefly described the research methods, especially the questionnaire used, and also briefly referred to its validation
6. as many as 19 of the surveyed women were> 25 years old, such a distribution does not reflect the demographic status in Africa
7. as the authors of the research rightly point out, women were poorly educated. It is true that the obtained results are consistent with the discharge in African societies, but they have a significant influence on the obtained results
8. women did not use contraception, did not use folic acid and the vast majority did not plan pregnancy at all. I understand that misums describe reality as it is, but the results are terrifying. It is not the fault of the medical community, but many independent factors.
9. the question arises - what useful information will readers from other regions of the world obtain from this publication?
Author Response
Thanks for the review. Please find the attached document
